# Factors Associated with Outpatient Satisfaction in Provincial Tertiary Hospitals in Nanchang, China: A Structural Equation Modeling Approach

**DOI:** 10.3390/ijerph19148226

**Published:** 2022-07-06

**Authors:** Xiaojun Zhou, Qiuwen He, Qi Li, Jie Kuang, Yalan Han, Jiayan Chen

**Affiliations:** 1Jiangxi Province Key Laboratory of Preventive Medicine, School of Public Health, Nanchang University, Nanchang 330006, China; zhouxiaojun@ncu.edu.cn (X.Z.); he1837024809@163.com (Q.H.); liqilq@ncu.edu.cn (Q.L.); kuangjie@ncu.edu.cn (J.K.); 2Library of Nanchang University, Nanchang University, Nanchang 330006, China; hanyalan@ncu.edu.cn

**Keywords:** outpatient satisfaction, tertiary hospital, structural equation model, related factors, China

## Abstract

Outpatient satisfaction is important in evaluating the performance of tertiary public hospitals in China. However, only a few studies have examined the interaction between outpatient satisfaction and its related factors. This study aimed to explore the relationship between patient satisfaction and its related factors in provincial tertiary hospitals. Six hundred outpatients in three provincial tertiary hospitals in Nanchang, China, were randomly selected. Structural equation modeling was used to analyze the relationship of the factors associated with outpatient satisfaction. The conceptual model fitted the data well (*χ*^2^/*df* = 4.367, CFI = 0.951, TLI = 0.937, SRMR = 0.055, RMSEA = 0.075), with all the path coefficients being statistically significant (*p* < 0.001). The environment and facilities showed the most significant influence on outpatient satisfaction (standardized total effect = 0.389), followed by the quality of diagnosis and treatment (standardized total effect = 0.235). The waiting time for medical services showed a partial mediation effect of 0.077 between the environment and facilities and outpatient satisfaction. The study indicates that targeted measures should be taken to improve the amenities of hospitals and shorten the waiting time for medical services, thus further improving outpatients’ medical experience.

## 1. Introduction

In China, the health administration organizes three-tier medical institutions to deliver healthcare services to rural and urban residents. The medical institutions are managed in a hierarchical model composed of primary, secondary, and tertiary hospitals according to their functions and tasks [1,2]. The tertiary hospitals (including national, provincial, municipal, and university-affiliated hospitals) are the highest level of medical institutions in China; they provide high levels of general and specialist medical services and undertake the tasks of clinical teaching, training, and scientific research [3]. Owing to the unequal allocation of medical resources, most patients in China tend to visit tertiary hospitals; this has overloaded the hospitals and increased concerns about medical services [4]. To address the issues of medical services supplied further, the general office of the State Council of China promulgated the *Opinions on Strengthening the Performance Evaluation of Tertiary Public Hospitals* in 2019 [5]. Patient satisfaction was included as an important indicator of the tertiary public hospitals’ performance evaluation index system [5]. The performance assessment results reported by the National Health Commission of China indicated that the patients were not satisfied with outpatient services, such as the environment and facilities of the hospital and doctor–patient communication [6]. Therefore, the issues regarding outpatient services need to be addressed.

Although previous studies have typically concentrated on reporting patient satisfaction and its related factors [7,8], only a few studies have systematically examined the association between them. For example, Ren and colleagues examined the effects of waiting time, doctor–patient communication, professional services, and accessibility for treatment information on patient satisfaction in hospitals in Henan, China [9]. However, considering the differences in socioeconomic levels in different areas of China, the findings may be limited in generalization. At present, further research is needed to explore the relative importance of the various factors related to outpatient satisfaction with tertiary hospitals in China and the actual relationship between them.

In this context, this study aimed to employ a structural equation modeling (SEM) approach that evaluates the relationship between outpatient satisfaction and its related factors. Since this is one of the first studies to undertake an SEM analysis of outpatient satisfaction with tertiary hospitals in China, it is hoped that this research will enhance our understanding of outpatient satisfaction improvement.

### 1.1. Relationship between Waiting Time for Medical Services, Environment and Facilities, and Outpatient Satisfaction

Recent evidence suggests that the waiting time for medical services impacts outpatient satisfaction [10,11,12]. For example, most tertiary hospitals in China have experienced a massive influx of patients, with up to 20,000 daily outpatient visits [4,13]. This massive influx of patients places the tertiary hospitals under extreme tension, leading to extended waiting times and inadequate doctor–patient communication, reducing the outpatients’ perceived satisfaction with visits [11]. Although methods such as improving appointment systems have been used to reduce the waiting time, outpatients still face long waits due to the imbalance between supply and demand for medical services [11]. The long waits may lead to inadequate treatment and poor doctor–patient communication, provoke the occurrence of doctor–patient conflicts, and result in dissatisfaction [14]. Related research has confirmed the prevalence of this phenomenon in developing countries [15].

The environment and facilities of the hospital were reported to have a close association with the waiting time for medical services [16]. Previous studies have confirmed that well-organized amenities such as clear signage settings, appropriate spatial layout, and guidance services could effectively decrease the waiting time [17]. In addition, the environment and facilities have been found to have a close relationship with outpatient satisfaction [18,19]. The outpatients’ first impression of the hospital starts with the outpatient setting. A noisy, crowded, and uncomfortable outpatient environment generates negative feelings and low satisfaction [18,20].

### 1.2. Relationship between Quality of Diagnosis and Treatment, Costs of Diagnosis and Treatment, and Outpatient Satisfaction

The quality of diagnosis and treatment is essential to outpatient satisfaction [21,22]. It is found that patients in China are most concerned with the quality of medical service issues [23]. When the patients believe their health status will improve after receiving the medical treatment, their trust and satisfaction with the hospital or the practitioner will be enhanced [24,25]. Previous studies have indicated the crucial impacts of diagnosis and treatment quality on patient satisfaction in China [23,26].

The costs of diagnosis and treatment are also related to outpatient satisfaction [27]. Several studies have proved that the higher costs of diagnosis and treatment resulted in lower patient satisfaction [28,29]. It is indicated that the diagnosis and treatment costs are associated with patients’ treatment expectancy. If patients expect too much from the effects of diagnosis and treatment, they may perceive that the medical treatment is not as effective as expected and claim that the fees paid for the medical services are not that worthy, which results in strong dissatisfaction with the hospital [30]. In China, several studies demonstrated that medical fees have been a salient concern for outpatients and were one of the critical influences on outpatient satisfaction [27,31].

Furthermore, the diagnosis and treatment costs may be affected by the quality of diagnosis and treatment [32]. The profit-seeking behavior of the medical institution and information asymmetry may lead to excessive examination and over-treatment, which are a constant feature of some hospitals, significantly increasing the patients’ out-of-pocket costs [31,33]. As a result, significant financial burden is imposed on patients and causes high patient dissatisfaction [7].

### 1.3. Theoretical Hypothesis

Taken together, this study has the following hypothesis:

**Hypothesis** **1** **(H1).***The environment and facilities, the quality of diagnosis and treatment, the waiting time for medical services, and the diagnosis and treatment costs directly affected outpatient satisfaction*.

**Hypothesis** **2** **(H2).***The waiting time for medical services mediated the environment and facilities and outpatient satisfaction*.

**Hypothesis** **3** **(H3).***The diagnosis and treatment costs mediated the quality of diagnosis and treatment and outpatient satisfaction*.

## 2. Materials and Methods

### 2.1. Participants

Outpatients from provincial tertiary hospitals in Nanchang, China, were selected. The outpatients refer to those who completed the medical treatment process at the selected hospital and provided informed consent at the time of the survey. Patients using other services (emergency, special needs, inpatient, health check-ups, and consultation) were excluded. A simple random sampling method was applied. Three hospitals were randomly selected from the twelve provincial tertiary hospitals in Nanchang City. The outpatients were selected in each hospital using an equal-volume sampling method (200 participants from each).

### 2.2. Survey Methods

The survey was conducted from January to February 2021, based on the *Third-Party Evaluation of the National Action Plan for Improving Medical Services Further* [34]. It is noted that this survey was conducted during the coronavirus disease 2019 (COVID-19) pandemic. However, due to China’s effective prevention and control activities, no confirmed COVID-19 cases were reported in Nanchang city during the study. Therefore, the present study did not include variables related to the pandemic.

With the coordination of the hospitals, the third-party investigators who had undergone unified training performed the survey. The investigators stayed in the outpatient pharmacy and distributed the questionnaire to the outpatients, who completed the medical treatment process. A systematic sampling approach was used. The participants were selected every five intervals (k = 5) according to their outpatient sequential number until the expected sample size of each hospital was achieved. If the selected outpatient refused to participate, the next patient was administered the questionnaire. Before distributing the questionnaire, the investigators briefly explained the purpose, significance, and content of the survey and obtained written informed consent from the participants. Then, the participants were asked to complete an anonymous and self-administered questionnaire. After returning the questionnaire, the investigators checked if there were missing or incorrect responses and requested the participants to add or correct the information. Finally, 662 patients were investigated, and 62 of them declined the survey (90.63% response rate). Six hundred valid questionnaires were returned, an effective recovery rate of 100%.

### 2.3. Survey Contents

This survey was adapted from the questionnaire used in the *Third-Party Evaluation of the National Action Plan for Improving Medical Services Further* [34]. It included two main sections: (1) basic information about patients, such as gender, age, occupation, education level, and type of medical insurance; (2) information on outpatient satisfaction, including the *environment and facilities* (adequate seats in the waiting room, clean water in the waiting room, clear signage settings, appropriate spatial layout, and guidance/self-services), *quality of diagnosis and treatment* (communication about patients’ condition, explanation of medical examination reports, explanation of treatment protocols, over-diagnosis and over-treatment, respect and comfort, and respect for confidentiality), *waiting time for medical services*, *diagnosis and treatment costs*, and *overall satisfaction*. The Likert scale scoring method (0–5) was used to evaluate the outpatient satisfaction. Higher scores indicated a greater degree of satisfaction. The questionnaire used in this study is presented in the Appendix A.

### 2.4. Statistical Analysis

Continuous variables were described as the mean and standard deviation, and the categorical variable was shown as the frequency and constituent ratio. The outpatient satisfaction differences in participant characteristics were compared using the *t*-test or analysis of variance. The conceptual model was constructed and evaluated using SEM. In order to determine whether the items (observed variables) could reflect the corresponding two latent variables (the *environment and facilities* and *quality of diagnosis and treatment*), an exploratory factor analysis (EFA) was conducted. The principal component analysis and varimax rotation were performed in EFA. The criterion of eigenvalue >1 was used for retaining the number of components. The SEM was applied to analyze the relationship between outpatient satisfaction and its related factors. The maximum likelihood estimation was performed in the SEM. The ratio of chi-square to degrees of freedom (*χ*^2^/*df*), comparative fit index (CFI), Tucker–Lewis index (TLI), standardized root mean square residual (SRMR), and root mean square error of approximation (RMSEA) were reported. The acceptable fit index criteria used in SEM were as follows: *χ*^2^/*df* < 5.00, CFI > 0.90, TLI > 0.90, SRMR < 0.08, and RMSEA < 0.10. The significance level was set at *α* = 0.05 (two-tailed).

## 3. Results

### 3.1. Characteristics and Outpatient Satisfaction of the Participants

The characteristics of the participants are shown in Table 1. The participants’ mean age was 38.69 years. The majority were female (72.83%), local residents (85.00%), with an urban household registration (67.83%). Nearly half (49.50%) had undergraduate or junior college education. In addition, 33.00% of the outpatients were enterprise employees, 43.00% reported annual household incomes of less than CNY 60,000, and 42.83% had resident medical insurance. The overall outpatient satisfaction was 4.62 ± 0.62. No significant differences in satisfaction were observed in the characteristics.

### 3.2. Validation of the Measurement Models

For validation of the measurement models, 55 outpatients were selected. The KMO measure of sampling adequacy (0.834) and Bartlett’s test of sphericity (*χ*^2^ = 486.22, *p* < 0.001) indicated an appropriate correlation matrix for EFA. A two-factor structure with each eigenvalue >1 (6.21 and 1.45) was shown in Table 2. The two-factor solution accounted for 69.63% of the total variance. Factor 1 showed greater loadings on six items, reflecting the quality of diagnosis and treatment. Factor 2 showed more significant loadings on the other five items, which indicated the environment and facilities of the hospital. Cronbach α coefficients for the two factors were 0.915 and 0.802, suggesting good internal consistencies of the two measurement models. It is noted that several items (such as adequate seats in the waiting room) were suspected to be cross-loading. Their loadings in both Factor 1 and Factor 2 were >0.40. We tried to remove these items and found that deleting the items hardly changed the factor structure, but reduced the internal consistency of the questionnaire. Since previous research has demonstrated that an item could be assigned to the common factor that reported the highest loading [35], the items were retained in the questionnaire.

### 3.3. Relationship between Outpatient Satisfaction and Its Related Factors Based on SEM

The initial model was constructed according to the theoretical hypothesis put forward in the introduction (Figure 1). The initial model showed an acceptable fit to the sample data (*χ*^2^/*df* = 4.789, CFI = 0.934, TLI = 0.918, SRMR = 0.056, RMSEA = 0.080). However, the two path coefficients denoted non-significant relationships (quality of diagnosis and treatment → diagnosis and treatment costs, *p* = 1.000; diagnosis and treatment costs → outpatient satisfaction, *p* = 0.542), suggesting that the initial model needed to be modified.

We removed the indicator (diagnosis and treatment costs) that had a non-significant relationship with other variables. The corrected model (Figure 2) had a good fit (*χ*^2^/*df* = 4.367, CFI = 0.951, TLI = 0.937, SRMR = 0.055, RMSEA = 0.075) with all path coefficients being statistically significant (*p* < 0.001).

### 3.4. Effects of the Factors on the Outpatient Satisfaction Based on SEM

As shown in Table 3, both the quality of diagnosis and treatment and the waiting time for medical services directly affected outpatient satisfaction, with the standardized direct effects of 0.235 and 0.153. The environment and facilities, directly and indirectly, affected outpatient satisfaction, with a total effect of 0.389. The waiting time for medical services partially mediated the environment and facilities and the outpatient satisfaction, with a mediation effect of 0.077, accounting for 19.79% of the total effect (0.389).

## 4. Discussion

This study found an outpatient satisfaction of 4.62 ± 0.62 in China, which was higher than observed in previous studies [23,36]. This finding might be explained by the fact that the outpatient service management (such as the medical service process, rational use of medicine, level of medical technology, and internal management of health facilities) has improved in China in recent years [6]. These considerable efforts made by the Chinese government have created favorable conditions for meeting the outpatients’ health needs, thus increasing their overall satisfaction with outpatient services.

Using an SEM model, our study also evaluates the relationship between outpatient satisfaction and its related factors. The results indicate that the environment and facilities, the quality of diagnosis and treatment, and the waiting time for medical services are directly associated with outpatient satisfaction with provincial tertiary hospitals in China, partly in line with our first hypothesis. In detail, the environment and facilities have the most significant influence on outpatient satisfaction, followed by the quality of diagnosis and treatment.

Numerous studies have confirmed the positive impact of the environment and facilities on outpatient satisfaction [18,37]. Similar to these studies, our findings further support the significance of environment and facilities in outpatient satisfaction. In our study, the environment and facilities were attributed to clear signage settings, appropriate spatial layout, adequate seats in the waiting room, clean water in the waiting room, and guidance/self-service. The clear signage settings are essential in optimizing the outpatients’ triage [38]. The appropriate spatial layout could reduce ineffective movement of the outpatients [17]. Adequate seats and clean water in the waiting room create a comfortable environment, which may alleviate the outpatients’ negative emotions and avoid the chaos caused by the long waiting time [18,37,39]. In addition, practical guidance or self-services helps select the appropriate departments [39]. Based on our findings, promoting the services above is particularly important for improving the outpatients’ health-seeking experiences.

Another finding in our study is that the quality of diagnosis and treatment directly affects outpatient satisfaction. It is consistent with that of Zhou and colleagues [21]. First, the primary purpose of seeking medical service for most outpatients in China is to be cured [4,14]. However, most of them lack medical expertise. The information asymmetry may contribute to the tendency toward dependence on clinicians [31]. Therefore, promoting patient–doctor communication and having a patient and meticulous attitude to explaining the medical examination reports and treatment protocols may improve the outpatients’ satisfaction [13,40]. Second, with the changes in health needs, humanistic care, such as respect, comfort, and confidentiality, receives growing attention from the outpatients [4]. Therefore, enhancing the non-medical aspects of healthcare is vital in improving the outpatients’ satisfaction.

The results of the SEM also demonstrated that the waiting time for medical services is a relevant factor for outpatient satisfaction, supporting evidence from Ma and colleagues [11]. This finding could be attributed to a universal phenomenon called “three long one short” (long queues for registration, medical examination, payment, and a short time for diagnosis and treatment) in the tertiary public hospitals in China [14,39]. When the waiting time is longer than expected, the patients may be prone to negative emotions, affecting their satisfaction [41]. One interesting finding in our study is that the waiting time for medical services partially mediated the relationship between the environment and facilities and outpatient satisfaction, which agrees with our second hypothesis and is consistent with the findings of Sun and colleagues [42]. It has been suggested that both the appropriate spatial layout and guidance service supplied contribute to shortening the waiting time, improving the outpatients’ satisfaction [16]. In this regard, the tertiary public hospitals should strengthen the construction of amenities and optimize triage of the outpatients, shortening their waiting time. On the other hand, improving the outpatient appointment service and rationalizing the visit schedule of patients may also help improve outpatient satisfaction [19].

Although we assumed that the diagnosis and treatment costs play a role in outpatient satisfaction, no significant correlation was found. This result might be attributed to the implementation of policies in China, such as cancellation of the drug price addition, instruction of reasonable laboratory examinations [39], strict price regulation of medical services [7], the increased proportion of essential medicines in outpatient prescriptions [6], and the changing orientation of tertiary hospitals [43]. Implementing these policies promotes the rational use of drugs. This makes hospital charges more transparent, which may interfere with the association between the diagnosis and treatment costs and outpatient satisfaction.

Although we focus our study on outpatient satisfaction with provincial tertiary hospitals and present a better explanation of the association between outpatient satisfaction and its related factors based on SEM, several limitations need to be noted regarding the current study. First, the present study was conducted during the COVID-19 pandemic, therefore it can be argued that the medical resources supplied might be influenced by the prevention and control of the epidemic. Therefore, it might have an impact on the study outcomes. However, due to China’s effective prevention and control activities, no confirmed cases were reported in Jiangxi Province from March 2020 to October 2021. That is, during the implementation of the study, the participants were living a normal social life. Therefore, it is indicated that the effects of factors related to the COVID-19 pandemic on the study outcomes were limited. Second, since we aimed to investigate the outpatient satisfaction with provincial tertiary hospitals, the findings may be limited while generalizing to lower-level hospitals or clinics. Third, most of the outpatients in our study were from central China. Therefore, given the differences in sociodemographic characteristics, whether the findings based on the samples could be generalized to populations in other areas of China is unknown. Fourth, the present study mainly investigates the factors related to health-seeking experiences. Therefore, the role of other factors such as the severity of disease and emotions is unknown. Further investigation of these factors are needed. Finally, although the SEM approach was applied to assess the association between outpatient satisfaction and its related factors, the inference on causality was limited due to the cross-sectional study design.

## 5. Conclusions

The present study indicates that outpatient satisfaction with provincial tertiary hospitals in China is relatively high. The environment and facilities of hospitals show the most significant impact on outpatient satisfaction. Their correlation is partially mediated by the waiting time for medical services. The findings suggest that more attention should be paid to the non-health aspects of medical services while improving the quality of healthcare delivery. Therefore, the decision-makers should emphasize the outpatients’ health-seeking experiences and take targeted measures to improve the amenities of hospitals and shorten the waiting time for medical services.

## Figures and Tables

**Figure 1 ijerph-19-08226-f001:**
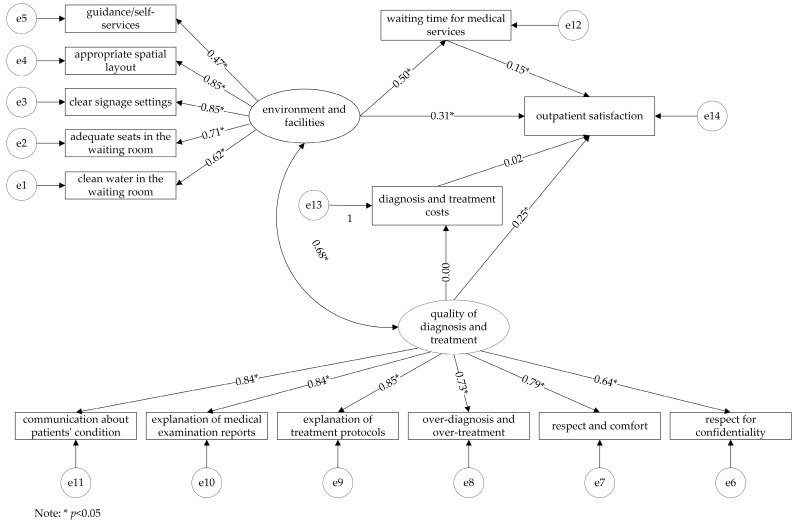
The initial model of outpatient satisfaction in provincial tertiary hospitals.

**Figure 2 ijerph-19-08226-f002:**
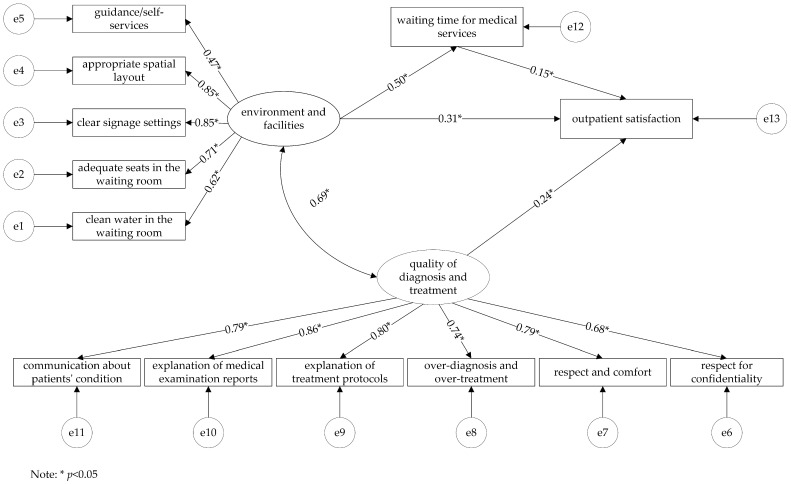
The corrected model of outpatient satisfaction in provincial tertiary hospitals.

**Table 1 ijerph-19-08226-t001:** Characteristics of the participants.

Characteristics		*N* (%)	Satisfaction (Mean ± SD ^1^)	*t*/*F*	*p*
Gender	Male	163 (27.17)	4.57 ± 0.59	−1.11	0.266
	Female	437 (72.83)	4.63 ± 0.63		
Age, years	<30	202 (33.67)	4.57 ± 0.68	1.17	0.311
	30~49	249 (41.50)	4.66 ± 0.58		
	≥50	149 (24.83)	4.61 ± 0.61		
Residence	Local resident	510 (85.00)	4.62 ± 0.62	0.46	0.645
	Non-local resident	90 (15.00)	4.59 ± 0.63		
Household registration	Rural	193 (32.17)	4.54 ± 0.71	−1.96	0.051
	Urban	407 (67.83)	4.65 ± 0.57		
Education	Junior high school and below	128 (21.33)	4.52 ± 0.70	2.06	0.105
	Senior high school/Technical secondary school/Technical school	150 (25.00)	4.71 ± 0.55		
	Undergraduate/Junior college	297 (49.50)	4.61 ± 0.62		
	Postgraduate and above	25 (4.17)	4.64 ± 0.57		
Occupation	Civil servant/State-owned enterprise employee	162 (27.00)	4.72 ± 0.53	1.94	0.086
	Private enterprise employee	127 (21.17)	4.62 ± 0.60		
	Retiree	44 (7.33)	4.48 ± 0.73		
	Farmer	32 (5.33)	4.44 ± 0.84		
	School student	47 (7.83)	4.64 ± 0.53		
	Others	188 (31.33)	4.59 ± 0.64		
Annual household income, CNY	<60,000	258 (43.00)	4.57 ± 0.62	1.57	0.209
	60,000~120,000	158 (26.33)	4.61 ± 0.68		
	≥120,000	184 (30.67)	4.68 ± 0.57		
Medical insurance	Employee medical insurance	205 (34.17)	4.64 ± 0.59	2.57	0.077
	Resident medical insurance	258 (43.00)	4.55 ± 0.69		
	others	137 (22.83)	4.69 ± 0.51		

^1^ Abbreviations: SD, standard deviation.

**Table 2 ijerph-19-08226-t002:** Validation of the measurement models.

Items of the Questionnaire	Factor 1	Factor 2
over-diagnosis and over-treatment	**0.937**	0.010
explanation of medical examination reports	**0.879**	0.271
respect for confidentiality	**0.770**	0.226
communication about patients’ condition	**0.768**	0.404
respect and comfort	**0.670**	0.496
explanation of treatment protocols	**0.669**	0.501
appropriate spatial layout	0.301	**0.796**
clean water in the waiting room	0.229	**0.795**
clear signage settings	0.293	**0.765**
guidance/self-service ^1^	0.072	**0.741**
adequate seats in the waiting room	0.492	**0.518**
Eigenvalue	6.21	1.45
% of the variance	56.48	13.15

Notes: Factors were extracted by principal component analysis and were rotated by varimax rotation. The highest factor-loading of each item was highlighted in bold. ^1^ Guidance service refers to consultation services (e.g., informing the patients of the department’s location, recommending the department to visit) provided by guide-service staff. Self-service means that the patients use self-service machines equipped in the hospital for registration, payment, and retrieving physicians’ information.

**Table 3 ijerph-19-08226-t003:** Effects of the factors on the outpatient satisfaction based on SEM ^1^.

Indicators	Standardized Direct Effect	Standardized Indirect Effect	Standardized Total Effect
environment and facilities	0.312	0.077	0.389
quality of diagnosis and treatment	0.235	-	0.235
waiting time for medical services	0.153	-	0.153

^1^ Abbreviations: SEM, structural equation model.

## Data Availability

Data are made available by contacting the corresponding author and the first authors.

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
