# Peer review of "Factors Associated with Outpatient Satisfaction in Provincial Tertiary Hospitals in Nanchang, China: A Structural Equation Modeling Approach"

_ijerph, 2022, doi:10.3390/ijerph19148226_

Round 1
Reviewer 1 Report
This is an important study. However, this study needs some major revisions. Moreover, the following queries need to be clarified.
1. The relationships and selected variables are important in this study, the authors should describe their theoretical basis and hypotheses.
2. The questionnaire of this study is very important, and the authors should list the questionnaire in the appendix / supplementary material to let readers realize the detailed contents.
3. The methods need to be clearly written and explained so that all readers could recreate the study.
4. What are the limitations of this study? What do these limitations affect the results or conclusions?
5. Write simply, as it can be a beautiful thing if many readers can understand your research story.
Reviewer 2 Report
Dear authors,
e topic of your paper is very interesting and certainly topical, but some adjustments seem to me necessary to improve the understanding of readers, unfamiliar with the Chinese healthcare system. Furthermore, both the description of the method and the results should be better explained.
Below is the file with some of my suggestions:
Introduction
line 33: I think it is necessary to explain to readers unfamiliar with the Chinese healthcare system, what tertiary services are, even if briefly
Lines 53-63: the explanations given in lines 53-63 concerning the different links between quality of diagnosis, examination costs, patient satisfaction should be better explained
2.2. Survey methods
In this section, the authors describe a data collection carried out during the COVID-19 pandemic, but no reference is made to this. Perhaps a justification concerning the reasons why they did not take this variable into account should be added here and, above all, added in the research limitations.
Lines 100-101 “A face-to-face interview was taken using an anonymous questionnaire”. This statement sounds a bit strange. How can you do a face-to-face interview using a questionnaire in an anonymous form. What does this mean? If it is an interview, the interviewee is not anonymous.
The questionnaire is not adequately described.
2.3. Survey contents
Line 111: what do you mean with guidance/self-services ?
3.2. Validation of the measurement models
Table 2: “communication about patients’ condition”, and even more, “respect and comfort” and “the explanation of treatment protocols” load on the two factors with a saturation value higher than 0.4. The same thing happens for “adequate seats in the waiting room” that load both in F2 (0.518) and in F1 (0.492). Some explanatory notes should be added.
Discussion
Much of the literature cited by the authors in support of their results could also have been cited in the introduction. The reader would certainly benefit from this. Understanding would also be facilitated with respect to the objectives of the research itself, which should be reiterated
Round 2
Reviewer 1 Report
This study is worthy to be published.